# PAX1/JAM3 Methylation and HPV Viral Load in Women with Persistent HPV Infection

**DOI:** 10.3390/cancers16071430

**Published:** 2024-04-07

**Authors:** Mingzhu Li, Chao Zhao, Xiaobo Zhang, Jingran Li, Yun Zhao, Wei Zhang, Lihua Ren, Lihui Wei

**Affiliations:** Department of Obstetrics and Gynecology, Peking University People’s Hospital, No. 11 Xizhimen South Street, Beijing 100044, China; mingzhu1815@bjmu.edu.cn (M.L.); 0062034740@bjmu.edu.cn (C.Z.); zhangxiaobo@pkuph.edu.cn (X.Z.); lijingran@pkuph.edu.cn (J.L.); zhaoyun@pkuph.edu.cn (Y.Z.); zhangwei@pkuph.edu.cn (W.Z.); renlihua@pkuph.edu.cn (L.R.)

**Keywords:** methylation, human papillomavirus (HPV), viral load, junctional adhesion molecule (JAM3), paired box gene1 (PAX1), persistent

## Abstract

**Simple Summary:**

Persistent infection with high-risk HPV is the main cause of cervical cancer. However, the changes in epigenetics and viral load (VL) during persistent HPV infection are not well understood. This study selected individuals with a persistent HPV infection but without developing high-grade cervical lesions, analyzed the changes in PAX1/JAM3 methylation and VL according to the duration of HPV infection, and found that in women with HPV infections persisting for more than 3 years, there is a notable increase in the methylation levels of PAX1/JAM3, which could be used as cumulative evidence of persistent HPV infection before the occurrence of precancerous lesions. HPV infection persisting for more than 3 years is more likely to be associated with vaginal lesions, and HPV VL could be used as an indicative biomarker for concurrent cervical–vaginal lesions, which might be helpful to provide a clinical perspective for monitoring and intervention of individuals with persistent HPV infection.

**Abstract:**

The relationship of PAX1/JAM3 methylation as well as HPV viral load (VL) with cervical lesions has been reported, but their role in persistent HPV infection without cervical high-grade lesions has not been fully elucidated. A total of 231 females diagnosed with persistent HPV infection and pathologically confirmed absence of high-grade cervical lesions were selected from the Colposcopy Outpatient Clinic of Peking University People’s Hospital, from March 2023 to December 2023. They were categorized into two groups based on the duration of HPV infection: the HPV persistent less than 3 years group and the more than 3 years group. PAX1/JAM3 methylation and HPV VL were determined by real-time PCR and BioPerfectus Multiplex Real-Time (BMRT)-HPV reports type-specific VL/10,000 cells, respectively. The average age of individuals with HPV infection lasting more than 3 years was higher compared to those with less than 3 years (48.9 vs. 45.1 years), with a statistically significant difference. Among the participants, 81.8% (189/231) had no previous screening. The methylation levels of JAM3 and PAX1 were significantly higher in individuals with HPV infection persisting for more than 3 years compared to those with less than 3 years, with a statistically significant difference (*p* < 0.05). There was a significant correlation between PAX1 and JAM3 methylation (*p* < 0.001), which could be used as cumulative evidence of HPV infection duration before the occurrence of precancerous lesions. The incidence of vaginal intraepithelial lesions was higher in individuals with HPV infection persisting for more than 3 years compared to those with less than 3 years, and HPV VL can be used as an indicative biomarker for concurrent cervical–vaginal lesions, especially for HPV other than 16/18 genotypes.

## 1. Background

Human papillomavirus (HPV) is a small, non-enveloped, double-stranded circular DNA virus, approximately 8 kb, infecting the basal keratinocytes of mucosal and cutaneous epithelia [1]. Although HPV encodes only 8 genes, their impact extends across numerous cellular pathways within host target cells. HPVs have evolved mechanisms to efficiently replicate viral DNA, suppress host responses to viral DNA, and establish an immune-privileged microenvironment to ensure successful viral replication and the production of viral particles [2]. 

There are over 200 identified HPV types, and fifteen HPV types were classified as high-risk types (16, 18, 31, 33, 35, 39, 45, 51, 52, 56, 58, 59, 68, 73, and 82); 3 were classified as probable high-risk types (26, 53, and 66); and 12 were classified as low-risk types (6, 11, 40, 42, 43, 44, 54, 61, 70, 72, 81, and CP6108), reported by Munos et al. in 2003 [3]. In 2012, the International Agency for Research on Cancer (IARC) reported that high-risk carcinogenic HPV genotypes, typically including up to 14 types (HPV16, 18, 31, 33, 35, 39, 45, 51, 52, 56, 58, and 59, which are Group 1 carcinogens, and HPV66 and 68, which are considered possibly/probably carcinogenic for humans (Group 2A/B), have been included in the 2021 WHO guidelines [4]. Persistent high-risk HPV (hrHPV) infection is associated with the development of cervical precancerous lesions and cervical cancer [5]. HPV16/18 are the highest-risk genotypes, and distinguishing them from other hrHPV genotypes is recommended. Although most HPV infections are transient and can spontaneously be cleared within 1 or 2 years, approximately 10–20% of infections persist latently, leading to disease progression [6]. Several factors, including host factors (age, cervicovaginal microbiome, immunity, oral contraception, smoking) and viral factors (genotype, variants, viral load, integration...), etc., [7,8,9,10,11] have been reported as common high-risk factors for causing persistent HPV infection. 

A new infection places the woman at a lower risk for cervical intraepithelial neoplasia grade 2 or worse (CIN2+) than a persistent infection with the same genotype [12]. However, women with HPV persistence will, in general, either become HPV-negative or develop CIN2+ within 6 years, even with intensive clinical follow-up [13]. However, from the time of HPV infection until the onset of precancerous cervical lesions, there are no other well-established indicators for monitoring other than repeated HPV tests and referral for colposcopy to exclude cervical lesions. Whether there are subtle changes in other biomarkers during the long history of HPV infection has not been extensively reported.

It is known that cytology is recommended as the routine triage method for HPV-positive women [14]. The low sensitivity of cytology means that some hrHPV-positive women may have CIN2+ despite normal cytology in the triage test. To increase the specificity of triage, some biological markers have recently been suggested, including the methylation of host or HPV genes [15], immunohistochemical detection of p16^INK4a^ in liquid-based cytology [16], and the use of HPV viral load [17] et al.

HPV viral load (VL) has been studied more recently and has been reported to have the ability to predict the outcome of an hrHPV infection and the grade of cervical lesions [18,19]. It has been studied for triaging of hrHPV-positive women. High-risk HPV VL has shown a specificity of 96.4% and a sensitivity of 88% in distinguishing between women with high- and low-grade abnormal cervical cytology [20]. However, variability in the literature still exists regarding the correlation between VL and lesions. 

DNA methylation has also played an important role as a triage test in hrHPV-positive women, which is an attractive alternative to cytology, such as junctional adhesion molecule (JAM3), and is involved in processes such as leukocyte migration, blood vessel formation, and tumor metastasis [21]. Research suggests that for patients that are hrHPV positive, the methylation level of JAM3 shows higher sensitivity compared to liquid-based cytology (LBC), indicating that JAM3 methylation markers may serve as a novel biomarker guiding grading and treatment for hrHPV-positive patients [22]. 

The paired box gene1 (PAX1) is another potential methylation biomarker and is a transcription factor that plays crucial roles in diverse biological processes. It belongs to the pPAX gene family, which is highly conserved in both vertebrates and invertebrates [23]. Many studies have previously identified PAX1 as a methylation-silenced gene observed in cervical cancer. It can serve as an auxiliary biomarker for clinical examination of cervical cancer to improve the effectiveness of screening [24,25,26,27,28,29].

Previous studies have primarily focused on the correlation between methylation, HPV VL, and cervical precancerous lesions [30]. However, there is relatively limited research on the correlation of these two biomarkers in populations with early stage of persistent HPV infection before the occurrence of cervical lesions. This study aims to investigate the correlation between the duration of persistent HPV infection and methylation as well as HPV VL, providing objective evidence for the monitoring and subsequent intervention of persistent HPV infection.

## 2. Materials and Methods

### 2.1. Study Population and Samples

The patients with persistent HPV infection were recruited from the colposcopy outpatient clinic of Peking University People’s Hospital (PKUPU) from March 2023 to December 2023. Inclusion criteria were persistent HPV infection without confirmed cervical high-grade lesions or above (cervical intraepithelial neoplasia2+; CIN2+) by colposcopy-guided biopsy histopathology. Exclusion criteria included cytology indicating high-risk populations [atypical squamous cells cannot exclude HSIL (ASC-H), high-grade squamous intraepithelial lesion (HSIL), atypical glandular cells (AGC), and above]; imaging and physical examination suggesting cervical precancerous lesions or cervical cancer; previous cervical cancer patients; and a history of total hysterectomy. Persistent HPV infection was defined as the persistent infection with the same HPV genotype for 1 year or more [31,32]. HPV genotype change or unclear type were not considered. A total of 266 cases were enrolled in the study, with 23 cases confirmed as HSIL by colposcopy pathology and thus excluded. Two cases were removed due to high cytological risk (ASC-H, HSIL), and 10 cases were excluded due to missing specimens or incomplete information, leaving a final total of 231 cases included in the study (Figure 1).

### 2.2. Group and Previous History Collection

To determine the appropriate duration for grouping persistent HPV infection, we conducted stratified analyses for HPV infections for 1 year, 2 years, 3–4 years, and more than 5 years, and found that there were statistical changes in JAM3 methylation over 3 years (Figure 2). Therefore, 3 years of persistent HPV infection were selected as the critical value for grouping analysis.

The collection of medical history data mainly includes age, screening results (including cytology and HPV genotypes), and previous cervical cancer screening results. Researchers need to examine all medical records from the time of initial detection of HPV infection to the study date to determine the duration of HPV infection.

As the duration of persistent HPV infection is not an objective measure and the time of the first infection cannot be accurately determined, detailed records of the previous screening history are essential when collecting information from participants. This includes: (1) never screened before the initial detection of HPV infection; (2) previous screenings were negative before the initial detection of HPV infection (this refers to HPV-based testing; if previous screenings primarily relied on cytology, classified as the first category); (3) HPV was found to be positive before the initial detection of HPV infection (this mainly includes cases with different HPV types detected previously or cases that tested positive before but later turned negative), and the HPV duration of the latter was counted from recurrent positive to the study date; (4) have a history of treatment for HPV-related diseases (including continued positive HPV after treatment, and repeated positive HPV after negative follow-up). Previous HPV antiviral therapy was also recorded.

### 2.3. Methylation and HPV Viral Load

Before colposcopy, a disposable cervical brush (Hologic, Marlborough, MA, USA) was utilized to gather exfoliated cells from both the cervix and the transformation zone (TZ), and then the brush was placed into a 20 mL PreservCyt1 solution (Hologic, Marlborough, MA, USA) for testing. The samples were stored at 4 °C within 72 h and at −20 °C for long-term preservation. All samples were tested with cytology first, and the remaining preservation solution was used for methylation and HPV-VL testing. These tests are conducted within a certified DNA laboratory, with operators and staff members kept blinded to patients’ clinical information, cytology results, HPV genotyping, and cervical histopathology outcome. 

Genomic DNA (gDNA) was extracted from the cervical exfoliated sample utilizing the JH-DNA Isolation and Purifying kit (OriginPoly Bio-Tec Co., Ltd., Beijing, China) in accordance with the manufacturer’s guidelines. The DNA concentration was determined using the NanoDrop 2000c spectrophotometer (Thermo Fisher Scientific, Waltham, MA, USA). Subsequently, 200–1000 ng of gDNA underwent bisulfite conversion employing the JH-DNA Methylation-Lightning MagPrep kit (OriginPoly Bio-Tec Co., Ltd., Beijing, China) following the manufacturer’s protocol. Subsequently, the levels of *PAX1* methylation (*PAX1^m^*) and *JAM3* methylation (*JAM3^m^*) were determined using the “Human PAX1 and JAM3 gene methylation detection kit (real-time PCR)” for cervical cancer [Class III medical devices approved by the National Medical Products Administration (No. 20233400253)] with glyceraldehyde-3-phosphate dehydrogenase (GAPDH) as an internal control (OriginPoly Bio-Tec Co., Ltd., Beijing, China) by the SLAN-96S automatic medical PCR analysis system (Shanghai Hongshi Med Tech Co., Ltd., Shanghai, China) per the manufacturer’s instructions. The hypermethylation level of the *PAX1* gene or *JAM3* gene was determined by the difference between the two Ct values (ΔCt*_PAX1_* = Ct*_PAX1_* − Ct*_GAPDH_* or ΔCt*_JAM3_* = Ct*_JAM3_* − Ct*_GAPDH_*). The positive definition of *PAX1^m^* is Δ CtPAX1 ≤ 6.6, and *JAM3^m^* is defined as ΔCt*_JAM3_* ≤ 10.0. 

HPV genotyping and viral load were tested using BioPerfectus Multiplex Real-Time PCR (BMRT), which has been approved as sensitive as Cobas4800 for primary cervical cancer screening [33]; thus, we have opted to maintain its usage. The starting sample size for nucleic acid extraction is 300 μL, and the elution is 85 μL. The BMRT employs a PCR-based high-risk HPV assay utilizing a fluorescence-based multiplex HPV DNA genotyping kit (Bioperfectus Ltd., Taizhou, China). Specifically designed PCR primers and TaqMan probes were crafted for the 21 most prevalent HPV types targeting the HPV L1 gene, encompassing 14 high-risk HPV genotypes (HPV16, 18, 31, 33, 35, 39, 45, 51, 52, 56, 58, 59, 66, 68) as well as 7 medium- and low-risk HPV genotypes (HPV26, 53, 82, 73, 6, 11, 81). In this study, a 14-type high-risk BMRT assay was utilized. Additionally, a single-copy gene encoding DNA topoisomerase III (human TOP3) was amplified within the reaction to ensure DNA quality control and to determine the relative viral copy numbers in the samples. Normalization of HPV type-specific VLs was executed using the formula: VL = log10[(CnHPV/CnTOP3) × 10,000] copies/10,000 cells, where CnHPV represents the quantity of HPV DNA and CnTOP3 denotes the number of human cells. The experimental protocol closely followed the procedures outlined in previous reports [30,34].

### 2.4. Colposcopy and Pathology

Colposcopy-guided biopsies were performed in all women with persistent HPV infection to exclude precancerous cervical lesions. 2–4 biopsies were taken at any acetowhite area, and endocervical curettage (ECC) was performed in patients with type 3 TZ or suspicious cervical canal lesions. Additionally, the vaginal wall and fornices are examined for any lesions. Pathological diagnosis follows the 5th WHO classification of tumors, primarily including cases classified as negative for intraepithelial lesion or malignancy (NILM) and cervical intraepithelial neoplasia grade 1 (CIN1).

### 2.5. Statistical Methods 

Statistical analysis was conducted using SPSS 26.0 (IBM Corp., Armonk, NY, USA). The samples were stratified into two groups based on the duration of HPV infection. Descriptive statistics were employed to characterize the samples, with categorical variables presented as frequencies and percentages (e.g., cervical cytology, hrHPV genotype, histological and pathological results). Quantitative data were expressed as mean ± standard deviation (e.g., age), while non-normally distributed continuous variables were depicted as median and interquartile range (IQR), encompassing all HPV VL [log(all-hrHPV)], HPV16/18 VL [log(HPV16/18)], other 12 types of hrHPV VL [log(other-hrHPV)], *PAX1^m^*, and *JAM3^m^*. The Mann–Whitney U test was utilized to compare continuous variables between two independent groups. All analyses were two-sided, with statistical significance set at *p* < 0.05.

## 3. Result

In total, the average age of HPV infection was higher in 75 patients with infections lasting over 3 years than in 156 patients with infections lasting under 3 years (48.9 vs. 45.1), with a statistically significant difference. A total of 187 patients were infected with hrHPV genotypes. The top 7 persistent positive HPV genotypes were HPV16 (24.6%), HPV52 (23.5%), HPV58 (15.5%), HPV31 (12.8%), HPV56 (10.7%), HPV39 (9.6%), and HPV18 (8.6%), respectively. The top 4 HPV genotypes with the highest proportion of HPV duration of more than 3 years were HPV52 (17.7%), HPV16 (16.6%), HPV56 (11.1%), and HPV58 (10%), (Appendix A). One hundred and fifty-six had histologically normal tissues, and 75 had CIN1.

Among the 231 women, 81.8% (189/231) of the women had not been previously screened. The previous screening history, treatment history, antiviral treatment history, cytological results, persistent HPV types, and HPV VL showed no significant correlation with the duration of HPV infection. However, there was a statistically significant difference between *JAM3^m^* and *PAX1^m^* in HPV infection lasting more than 3 years versus less than 3 years (Figure 3). There were 28 cases of concurrent vaginal intraepithelial neoplasia (VaIN), and the incidence of concurrent vaginal wall lesions was higher in individuals with HPV infection lasting more than 3 years (Table 1).

The correlation analysis revealed a significant and highly correlated relationship between the methylation of PAX1 and JAM3 (*p* < 0.001). Additionally, there was a statistically significant correlation between JAM3 and total HPV VL (*p* = 0.037) but no significant correlation with genotypes (Figure 4) (Appendix A).

Analysis of individuals with concurrent VaIN revealed that there was a significant statistical difference in the overall HPV VL between those with and without vaginal lesions, primarily driven by HPV non-16/18 types, while no statistical difference was observed for HPV 16/18. There was no statistical difference observed in either type of methylation between individuals with and without concurrent VaIN (Figure 5).

## 4. Discussion

Unlike the previous study on the correlation between PAX1 methylation and HPV VL for the detection of cervical HSIL, this study explored the early changes before the occurrence of precancerous lesions from the correlation between the duration of HPV infection, methylation, and HPV VL. It was found that the average age of those with a persistent HPV infection lasting more than 3 years was higher than that of those with less than 3 years. Among prevalent infections, persistent HPV infections among older women were higher than those among younger age groups or new infections at any age [35]. In addition, PAX1 and JAM3 methylation changes occur when HPV persists for more than 3 years, especially JAM3 methylation. Interestingly, HPV VL did not correlate with the duration of HPV infection but was significantly associated with concurrent vaginal lesions.

Persistent hrHPV infections are considered to drive the progression of cervical dysplasia to invasive cervical cancer. Some carcinogenic HPV genotypes, such as HPV16, HPV18, HPV31, and HPV33, especially HPV16 persistence, have been reported to be associated with high absolute risks for progression to high-grade cervical lesions [36]. Women with persistent HPV16 infection have been reported to have a higher HPV VL at the initial screening test than women with transient infection [17] and could be used as a marker risk for potential progression to cervical precancerous lesions and invasive cancer [37].

Although some studies [19,38] have explored the correlation between HPV VL and disease severity, other findings proposed that HPV VL levels did not show any significant association with type-specific hrHPV persistence or subsequent development of cervical intraepithelial neoplasia, especially within women under 30 with normal cytology [39]. It seems that HPV16/18 VL plays a crucial role in contributing to HPV persistency [40] and was positively correlated with the grade of cervical lesions (CIN3 > CIN2 > CIN1, *p* < 0.05) [30]. However, it has also been found that lower HPV VL and viral genome integration predict the persistence of HPV16 but not the progression of a persistent HPV16 infection to CIN3+ in women with normal cytology [41]. Our study also found that HPV VL showed no significant correlation with the duration of HPV infection, regardless of HPV16/18 or other 12 genotypes. Considering that the population included in this study was mainly the population without precancerous lesions with persistent HPV infection, this period is an acute stage of HPV infection. As viral integration inhibits the replication and release of new virions, there exists an inverse correlation with circulating HPV VL. Consequently, as the cervical lesion progresses to CIN2-3, a notable decrease in HPV VL is observed due to the stable self-replication ability of HPV. However, upon progression from CIN2-3 to cervical cancer, the HPV gene integrates into the host cell’s DNA, leading to cell transformation. This integration results in the low initial HPV VL detected in cervical cancer tumors [42].

Overall, the relationship between HPV VL, persistent HPV infection, and oncogenicity remains to be explored. Some other factors can affect VL fluctuations, such as VL between sex partners, which is correlated and seems predictive of transmission episodes [43].

In addition, our study found that HPV persistence for more than 3 years is associated with a higher risk of concurrent vaginal lesions, which also explains why HPV positivity tends to persist because HPV-related diseases will not only cause cervical lesions but also genital–anal lesions. Without careful identification of lesions in the vagina, vulva, and perianal area during colposcopy, it is easy to miss a diagnosis. Additionally, we observed that high HPV VL was significantly associated with concurrent vaginal lesions, especially HPV non-16/18 types. It has been reported that concurrent genital–anal HPV often had significantly higher genital VL than genital-only HPV, and high HPV VL may facilitate genital–anal HPV concurrence [44]. 

Methylation showed a highly significant and increasing trend with disease severity. JAM3 methylation has been reported to show increased specificity and positive predictive value (PPV) for hrHPV-positive patients, surpassing cytology testing as a triage marker. Moreover, it demonstrates superior capability in distinguishing between productive and transforming CIN [21]. PAX1methylation exhibits its versatility, such as serving as a triage tool for detecting CIN3+ in women with positive HPV. Its clinical performance is comparable to cytology, with accuracy and specificity superior to HPV16/18 [45]. Moreover, PAX1 hypermethylation analysis may be preferable to hybrid capture (HC2) in triaging ASCUS and ASC-H [46]. Additionally, besides serving as an predictor for pathological upgrade of HSIL before cold knife conization [47], it may also predict the sensitivity and efficacy of concurrent chemo–radiotherapy (CCRT) in cervical cancer [48]. Furthermore, it may aid in the diagnosis of cervical adenocarcinomas [24].

JAM3 and PAX1 methylation have statistical significance with HPV sustained infection lasting more than 3 years compared to less than 3 years. Taking methylated JAM3 and PAX1 as another biomarker of sustained infection is a new finding in this study, suggesting that this group of women may progress more rapidly to precancerous lesions, which is worthy of clinical study. It may provide diagnostic value with respect to epigenetics, especially in cases of long-term HPV persistence where potential lesion development may occur.

Limitations: This study explores the duration of HPV infection as a determining factor in research. However, the duration of HPV infection obtained in this study is not based on prospective research but on cross-sectional research based on acquired previous screening history. Many of these histories were unclear, making it difficult to accurately determine the onset of HPV infection. Therefore, we documented the previous screening history and analyzed it, but we did not find any statistical differences between them. However, theoretically, the longer the duration of HPV infection, the higher the risk of developing precancerous lesions. Therefore, it is necessary to continue follow-up tracking in this population. In addition, due to the number of case limitations, it is difficult to analyze the correlation between methylation and HPV VL for each genotype in this research. Future samples will need to be expanded to explore the relationship between HPV VL and methylation among different HPV genotypes. In addition, the subjects of this study were women with persistent HPV infection without precancerous lesions, and colposcopy-guided biopsy indicated chronic inflammation, or CIN1, but this was the result of biopsy pathology, and the cases with hidden CIN2 could not be completely excluded. Therefore, the next plan of this study will focus on intervention with drugs for persistent HPV infection, continuing follow-up for one year to exclude the occurrence of CIN2+ during follow-up. Furthermore, the next step will focus on the role of the cervicovaginal microbiome in persistent HPV infection. The purpose of this study is mainly to explore methylation and HPV VL, and does not include these results.

In conclusion, with a persistent HPV infection lasting more than 3 years, there is a notable increase in the methylation levels of PAX1 and JAM3, which could be used as cumulative evidence of HPV infection duration before the occurrence of precancerous lesions. HPV infection persisting for more than 3 years is more likely to be associated with vaginal lesions, and HPV VL can be used as an indicative biomarker for concurrent cervical–vaginal lesions.

## Figures and Tables

**Figure 1 cancers-16-01430-f001:**
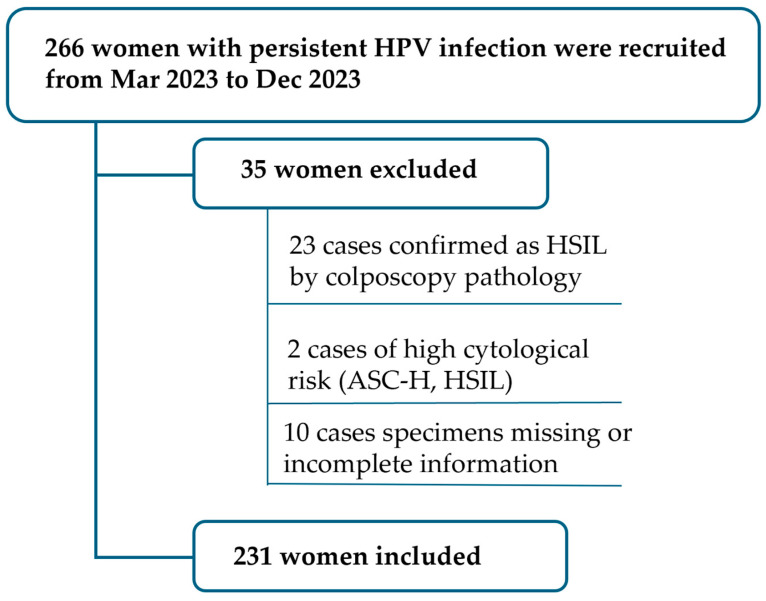
Flowchat of persistent HPV infection recruitment.

**Figure 2 cancers-16-01430-f002:**
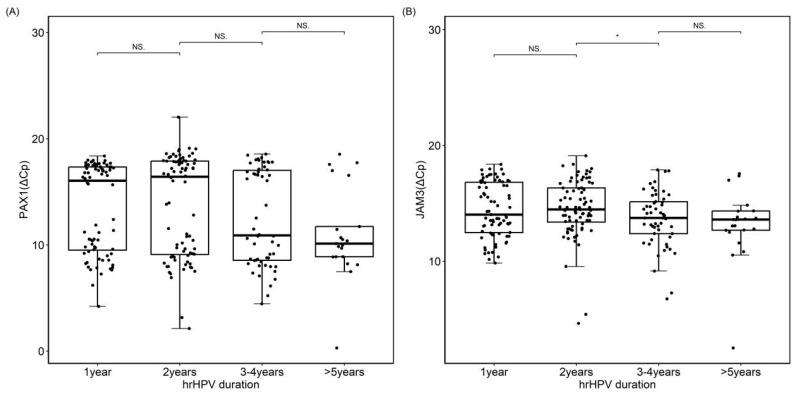
PAX1 and JAM3 methylation for different durations of HPV infection. (**A**). PAX1 methylation; (**B**). JAM3 methylation. NS: no significance; * indicate *p* < 0.05.

**Figure 3 cancers-16-01430-f003:**
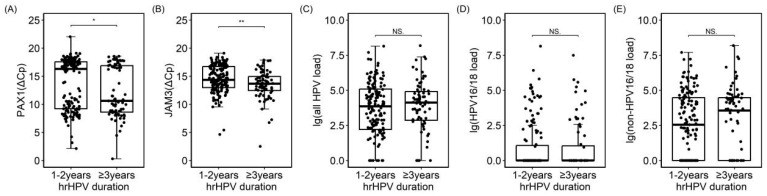
Dot plot of different tests and durations of HPV infection. (**A**). *PAX1 methylation*; (**B**). *JAM3 methylation;* (**C**). All HPV viral load; (**D**). HPV16/18 viral load; (**E**). Non–HPV16/18 viral load; *PAX1* (ΔCp) was defined as the Cp of methylated Paired Box Gene 1 gene–the Cp of internal control; *JAM3*, (ΔCp) was defined as the Cp of methylated Junctional Adhesion Molecule 3 gene–the Cp of internal control; non–HPV16/18 was defined as a positive hrHPV test result exclusive to HPV16/18; NS: no significance; * was mean *p* < 0.05, ** *p* < 0.01.

**Figure 4 cancers-16-01430-f004:**
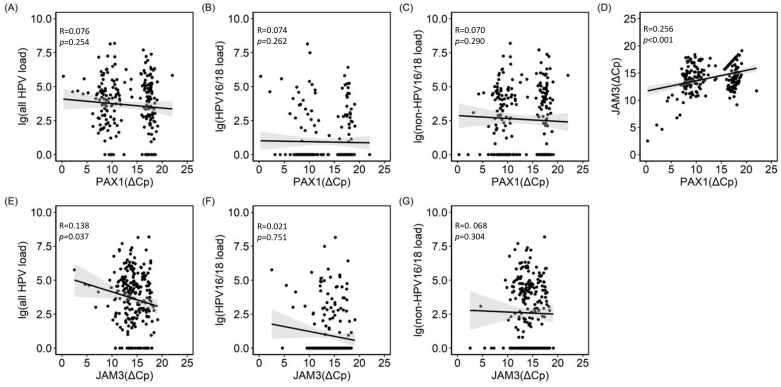
The relationship between HPV−VL and methylation tests. (**A**). *PAX1 methylation and* all HPV viral load; (**B**). *PAX1 methylation and* HPV16/18 viral load*;* (**C**). *PAX1 methylation and* non−HPV16/18 viral load; (**D**). *PAX1 methylation and JAM3 methylation*; (**E**). *JAM3 methylation and* all HPV viral load*;* (**F**). *JAM3 methylation and* HPV16/18 viral load; (**G**). *JAM3 methylation and* non−HPV16/18 viral load. PAX1 (ΔCp) was defined as the Cp of methylated Paired Box Gene 1 gene−the Cp of internal control; JAM3 (ΔCp) was defined as the Cp of methylated Junctional Adhesion Molecule 3 gene−the Cp of internal control; non−HPV16/18 was defined as a positive hrHPV test result exclusive to HPV16/18.

**Figure 5 cancers-16-01430-f005:**
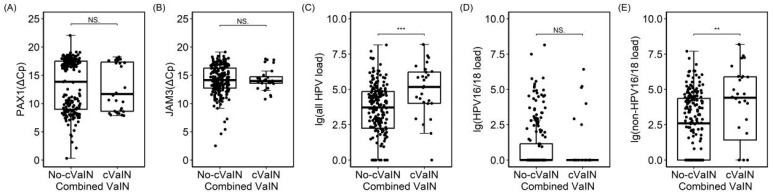
Dot plot of different tests in pathology normal/CIN1 and combined vaginal lesions. (**A**). *PAX1 methylation*; (**B**). *JAM3 methylation;* (**C**). All HPV viral load; (**D**). HPV16/18 viral load; (**E**). Non–HPV16/18 viral load; PAX1 (ΔCp) was defined as the Cp of methylated Paired Box Gene 1 gene−the Cp of internal control; JAM3, (ΔCp) was defined as the Cp of methylated Junctional Adhesion Molecule 3 gene−the Cp of internal control; non−HPV16/18 was defined as a positive hrHPV test result exclusive to HPV16/18; cVaIN, concurrent with vaginal intraepithelial neoplasia; NS: no significance; ** *p* < 0.01, and *** *p* < 0.001.

**Table 1 cancers-16-01430-t001:** Characteristics of Participants.

	Categories	Overall	HPV Duration		*p*
			1–2 Years	≥3 Years	
N		231	156	75	
Age (mean (±SD))		46.4 (12.3)	45.1 (12.4)	48.9 (11.8)	0.031
Previous Screening History (%)	No	189 (81.8)	130 (83.3)	59 (78.7)	0.261
	hrHPV (+)	22 (9.5)	15 (9.6)	7 (9.3)	
	hrHPV (−)	8 (3.5)	6 (3.8)	2 (2.7)	
	HSIL or CKC	12 (5.2)	5 (3.2)	7 (9.3)	
Cytology Results (%)	NILM	172 (74.5)	121 (77.6)	51 (68.0)	0.191
	ASC-US	36 (15.6)	23 (14.7)	13 (17.3)	
	LSIL	23 (10.0)	12 (7.7)	11 (14.7)	
HPV Testing Results (%)	non-hrHPV	44 (19.0)	31 (19.9)	13 (17.3)	0.899
	12 other hrHPV(+)	127 (55.0)	85 (54.5)	42 (56.0)	
	HPV16/18 (+)	60 (26.0)	40 (25.6)	20 (26.7)	
Anti-HPV drug history (%)	No	200 (86.6)	134 (85.9)	66 (88.0)	0.816
	Yes	31 (13.4)	22 (14.1)	9 (12.0)	
Log hrHPV VL (median [IQR])		4.0 [2.4, 5.0]	3.9 [2.2, 5.1]	4.1 [2.9, 4.9]	0.434
Log HPV16/18 VL (median [IQR])		0.0 [0.0, 1.1]	0.0 [0.0, 1.1]	0.0 [0.0, 1.0]	0.872
Log non-HPV 16/18(+) VL (median [IQR])		2.8 [0.0, 4.5]	2.5 [0.0, 4.5]	3.6 [0.0, 4.5]	0.556
*PAX1^m^* (ΔCp) (median [IQR])		12.8 [8.9, 17.5]	16.3 [9.2, 17.6]	10.6 [8.6, 16.9]	0.025
*JAM3^m^* (ΔCp) (median [IQR])		14.1 [12.7, 16.1]	14.4 [13.0, 16.7]	13.7 [12.4, 14.9]	0.007
Pathology (%)	Normal	156 (67.5)	104 (66.7)	52 (69.3)	0.799
	CIN1	75 (32.5)	52 (33.3)	23 (30.7)	
Concurrent VaIN (%)	VaIN 1	23 (10.0)	10 (6.4)	13 (17.3)	0.031
	VaIN 2/3	5 (2.2)	4 (2.6)	1 (1.3)	
	No VaIN	203 (87.9)	142 (91.0)	61 (81.3)	

HPV16/18 (+), positive results of HPV16 or HPV18; non−HPV16/18, positive hrHPV test result exclusive HPV16/18 positive; 12 other hrHPV (+), including HPV 12 genotype: 31, 33, 35, 39, 45, 51, 52, 56, 58, 59, 66, 68; non−hrHPV, HPV test result exclusive HPV 14 genotype: 16, 18, 31, 33, 35, 39, 45, 51, 52, 56, 58, 59, 66, 68. Abbreviations: N, numbers of subjects; SD, standard deviation; IQR: interquartile range; NILM, negative for intraepithelial lesion or malignancy; ASC-US, atypical squamous cells of undetermined significance; LSIL, low-grade squamous intraepithelial lesion; VL: viral load; CIN1, cervical intraepithelial neoplasm 1; VaIN: vaginal intraepithelial neoplasia; *PAX1^m^*, methylated Paired Box Gene 1 gene; *JAM3^m^*, methylated Junctional Adhesion Molecule 3 gene; (+), positive result; (−), negative result.

## Data Availability

The datasets generated and/or analyzed during the current study are available from the corresponding author upon reasonable request.

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
