# Peer review of "PAX1/JAM3 Methylation and HPV Viral Load in Women with Persistent HPV Infection"

_cancers, 2024, doi:10.3390/cancers16071430_

Round 1

Reviewer 1 Report

Comments and Suggestions for Authors

In this systematic article, authors aim to explore the correlation between PAX1/JAM3 methylation and HPV viral load (VL) in patients with persistent HPV infection without cervical high-grade lesions. Authors reported the results obtained from the analysis of cervical samples collected from women dividing the population into two groups based on the duration of HPV infection, less or more than 3 years of persistence.

The subject is very interesting because it is important to well define new biomarkers to be used as triage test for HPV DNA positive samples to discriminate persisting from transient infections.

The aim is clear but there are major revisions that require attention by the authors.

Simply Summary: Simply summary il lacking. Author should add this paragraph as indicated in the template of this journal.

Abstract: The abstract should be revised. As indicated in the template Authors should use the style of structured abstracts, but without headings. A short part of introduction should be also included.

The acronym BMRT should be specified.

Background: Authors reported in the manuscript: “There are over 200 identified HPV types and 14 HPV types (16, 18, 31, 33, 35, 39, 45, 51, 52, 56, 58, 59,66 and 68) are considered to be HR-HPV genotypes, of these, HPV16/18 are the highest risk genotypes, and distinguishing them from other HR-HPV genotypes is recommended by 2021 WHO guidelines”. 12 genotypes are considered carcinogenic for humans (Group 1) by International Agency for Research of Cancer (IARC). HPV66 and 68 are considered possible/probable carcinogenic for human (Group 2A/B). This is also reported in the Reference 2 (World Health Organization. WHO guideline for screening and treatment of cervical pre-cancer lesions for cervical cancer prevention, edition second.2021). This should be specified in this part of the manuscript.

 Materials and methods:

- Author reported that among inclusion criteria were HPV persistent infection without confirmed cervical high-grade lesions or above (CIN2+) by colposcopy-guided biopsy histopathology. However, in Table 2 of the Results paragraph is reported that 29 women resulted hrHPV negative. Why these women have been not excluded from the study? Author should better specify this result. The same suggestion is for the 2 women with ASC-H/HSIL. Authors reported that a cytology including ASC-H/HSIL should be an exclusion criteria. Moreover, it could be useful to add a chart to describe the study population.

- The criteria that the authors have used to define a persistent HPV infection < than 3 years or > than 3 years should be better explained. Have you collected the results of previous HPV testing of women enrolled? Among these persisting infections which hrHPV genotype resulted the prevalent?

- Authors should better describe which cytobrush was used and in which medium and which volume the cytobrush was resuspended

Authors should report the starting volume for nucleic acids extraction and the elution volume. This is an important pre-analytical information for the analytical evaluation of HPV viral load and for methylation.

Results: Table 2 should be revised. Authors should better define why both HPV negative women and women with HSIL lesion have been included in the analysis.

Comments on the Quality of English Language

Minor editing of English language required

Reviewer 2 Report

Comments and Suggestions for Authors

Comments to the Author:

Overall, the study is interesting and good settled. Besides the listed changes below, it needs some more corrections in English language.  

Introduction:

The first sentence is not good written. The second sentence: data to check; more HPV types are identified nowadays…

“As is well known, cytology is recommended” … replace with “It is known, … “.

The sentence: “DNA methylation has also played an important role as a triage test in HR-HPV-positive women, which is an attractive alternative to cytology, such as junctional adhesion molecule(JAM3), showed a higher specificity (88% vs 48%) and sensitivity (82% vs 91%) for detection of CIN3 as a triage test in HR-HPV-positive women[18] , comparing with cytology.” … should be split into two or refraised.

Viral load (VL) and other abbreviations should be correctly used afterwards all over in text.

Materials and methods:

Try to change some “included” and “excluded” words (the first chapter).

It is a bit questionable if the changes in JAM3 methylation over 3 years are reliable marker of persistent HPV infection (the second part).

Results:

The sentence: “The average age of HPV infection over 3 years was greater than that under 3 years (48.9 vs 45.1) in 77 and 156 patients, respectively, with a statistically significant difference.” … should be written correctly.

Discussion:

The same observations in this chapter:

Viral load (VL) and other abbreviations should be correctly used afterwards all over in text; … age of HPV infection … should be written correctly.

PAX1&JAM3 … to be corrected.

Comments on the Quality of English Language

Comments to the Author:

Overall, the study is interesting and good settled. Besides the listed changes below, it needs some more corrections in English language.  

Introduction:

The first sentence is not good written. The second sentence: data to check; more HPV types are identified nowadays…

“As is well known, cytology is recommended” … replace with “It is known, … “.

The sentence: “DNA methylation has also played an important role as a triage test in HR-HPV-positive women, which is an attractive alternative to cytology, such as junctional adhesion molecule(JAM3), showed a higher specificity (88% vs 48%) and sensitivity (82% vs 91%) for detection of CIN3 as a triage test in HR-HPV-positive women[18] , comparing with cytology.” … should be split into two or refraised.

Viral load (VL) and other abbreviations should be correctly used afterwards all over in text.

Materials and methods:

Try to change some “included” and “excluded” words (the first chapter).

It is a bit questionable if the changes in JAM3 methylation over 3 years are reliable marker of persistent HPV infection (the second part).

Results:

The sentence: “The average age of HPV infection over 3 years was greater than that under 3 years (48.9 vs 45.1) in 77 and 156 patients, respectively, with a statistically significant difference.” … should be written correctly.

Discussion:

The same observations in this chapter:

Viral load (VL) and other abbreviations should be correctly used afterwards all over in text; … age of HPV infection … should be written correctly.

PAX1&JAM3 … to be corrected.

Author Response

Overall, the study is interesting and good settled. Besides the listed changes below, it needs some more corrections in English language.  

Introduction:

The first sentence is not good written. The second sentence: data to check; more HPV types are identified nowadays…

Reply: Thank you for your suggestion. This part has been modified accordingly and different risk of HPV genotype have been added, but the 14 high-risk genotype recommended by WHO is mentioned.

“As is well known, cytology is recommended” … replace with “It is known, … “.

Reply: Has been revised.

The sentence: “DNA methylation has also played an important role as a triage test in HR-HPV-positive women, which is an attractive alternative to cytology, such as junctional adhesion molecule(JAM3), showed a higher specificity (88% vs 48%) and sensitivity (82% vs 91%) for detection of CIN3 as a triage test in HR-HPV-positive women[18] , comparing with cytology.” … should be split into two or refraised.

Reply: This part has been revised.

Viral load (VL) and other abbreviations should be correctly used afterwards all over in text.

Reply: all the abbreviations has been correctly used.

Materials and methods:

Try to change some “included” and “excluded” words (the first chapter).

Reply: Has been revised accordingly.

It is a bit questionable if the changes in JAM3 methylation over 3 years are reliable marker of persistent HPV infection (the second part).

Reply: Since there are currently few relevant reports on the duration of persistent HPV infection and changes in methylation, we divided it based on different durations of HPV infection, and preliminarily found that the methylation of JAM3 in HPV persistent infection for more than 3 years is significantly higher than that for less than 3 years. However, such findings need to be verified by future studies with larger samples

Results:

The sentence: “The average age of HPV infection over 3 years was greater than that under 3 years (48.9 vs 45.1) in 77 and 156 patients, respectively, with a statistically significant difference.” … should be written correctly.

Discussion:

The same observations in this chapter:

Viral load (VL) and other abbreviations should be correctly used afterwards all over in text; … age of HPV infection … should be written correctly.

PAX1&JAM3 … to be corrected.

Reply:Has been revised accordingly.

Reviewer 3 Report

Comments and Suggestions for Authors

In the manuscript entitled “PAX1/JAM3 Methylation and HPV Viral Load in Women with Persistent HPV Infection the authors present significant data concerning the association between HR-HPV persistent infection, the methylation status of PAX1, JAM3 genes and viral load. The manuscript is well written and data are sufficiently presented. However, some points need to be addressed.

Major concerns:

·         The introduction is required to be improved. In particular, authors should provide more details concerning the natural history of HPV infection. Τhe description of viral DNA structure and the molecular mechanisms involved in HPV induced carcinogenesis are required to be mentioned in order to help readers to better understand the biology of HPVs. Moreover, it is required to provide more information concerning the phylogenetic classification of HPV genotypes of Alpha - Papillomaviruses into High-Risk and Low-Risk genotypes, including their members and criteria that were used to classify the HPVs into these two categories (Expert Rev Mol Med. 2021;23:e19. doi:10.1017/erm.2021.18,  Virol. J. 2010;7:11.doi: 10.1186/1743-422X-7-11, Viruses. 2022 Dec 31;15(1):141. doi: 10.3390/v15010141).  Moreover, a short description of PAX1, JAM3 genes function would be essential in order to help readers to understand the impact of these two genes on cancer development.

·         In material and methods the authors need to make more clear which HPV DNA test was used for HPV genotyping. Moreover, the authors should describe in detail the methodology that was followed in order to determine the methylation status of  PAX1 and JAM3 genes.

·         In results the authors should describe the distribution of different HPV genotypes that were found in the examined specimens. More specific, BioPerfectus Multiplex Real-Time PCR (BMRT) assay has been developed to detect the different high risk and low risk HPV genotypes as well as their viral loads, simultaneously (BMC Cancer . 2015. doi.org/10.1186/s12885-015-1874-9). In the present analysis authors measure the total HPV DNA viral load, although high-risk HPVs comprise a group of 14 different HPV genotypes. Since different high risk HPV genotypes exhibit different distribution and carcinogenic capacity among the examined populations, I believe that the viral load of each genotype should be measured separately and subsequently these data should be further associated with the methylation status of PAX1/JAM3 genes. It would be interesting to investigate whether the viral load of specific high-risk HPV genotypes is associated with the methylation status of PAX1/JAM3 genes as well as with persistent infection in the examined patients.

Minor comments:

The analysis is focused on high-risk HPV genotypes. According to my opinion the title of the manuscript should be modified, thus emphasizing the high-risk HPV genotypes and persistent high-risk HPV infection.

Author Response

Review3:

In the manuscript entitled “PAX1/JAM3 Methylation and HPV Viral Load in Women with Persistent HPV Infection” the authors present significant data concerning the association between HR-HPV persistent infection, the methylation status of PAX1, JAM3 genes and viral load. The manuscript is well written and data are sufficiently presented. However, some points need to be addressed.

Major concerns:

  • The introduction is required to be improved. In particular, authors should provide more details concerning the natural history of HPV infection. Τhe description of viral DNA structure and the molecular mechanisms involved in HPV induced carcinogenesis are required to be mentioned in order to help readers to better understand the biology of HPVs. Moreover, it is required to provide more information concerning the phylogenetic classification of HPV genotypes of Alpha - Papillomaviruses into High-Risk and Low-Risk genotypes, including their members and criteria that were used to classify the HPVs into these two categories (Expert Rev Mol Med. 2021;23:e19. doi:10.1017/erm.2021.18,  Virol. J. 2010;7:11.doi: 10.1186/1743-422X-7-11, Viruses. 2022 Dec 31;15(1):141. doi: 10.3390/v15010141).Moreover, a short description of PAX1, JAM3 genes function would be essential in order to help readers to understand the impact of these two genes on cancer development.

 Reply: Thank you for your advice, we have added some introduction of the natural history of HPV infection.

  • In material and methods the authors need to make more clear which HPV DNA test was used for HPV genotyping. Moreover, the authors should describe in detail the methodology that was followed in order to determine the methylation status of  PAX1 and JAM3 genes.

 Reply: Thank you for your good question. HPV genotyping has also been conducted using BMRT, not only for HPV viral load but also HPV genotyping. We have added the related description for that.

  • In results the authors should describe the distribution of different HPV genotypes that were found in the examined specimens. More specific, BioPerfectus Multiplex Real-Time PCR (BMRT) assay has been developed to detect the different high risk and low risk HPV genotypes as well as their viral loads, simultaneously(BMC Cancer . 2015. doi.org/10.1186/s12885-015-1874-9). In the present analysis authors measure the total HPV DNA viral load, although high-risk HPVs comprise a group of 14 different HPV genotypes. Since different high risk HPV genotypes exhibit different distribution and carcinogenic capacity among the examined populations, I believe that the viral load of each genotype should be measured separately and subsequently these data should be further associated with the methylation status of PAX1/JAM3 genes. It would be interesting to investigate whether the viral load of specific high-risk HPV genotypes is associated with the methylation status of PAX1/JAM3 genes as well as with persistent infection in the examined patients.

 Reply: Thank you for your good recommendation. Statistical analysis was performed again and we added the distribution of the 14 high-risk HPV genotypes and their relationship with methylation as Supplementary table 1. Because of the number of cases limitation, if analyzed by each HPV genotype, the number of cases of each type is very small, it is difficult to analyze the exact correlation between the viral load of each genotype and methylation. But it’s what we going to do next, to expand the samples to explore the relationship between HPV viral load and methylation among different HPV genotypes, that is worth exploring.

Minor comments:

The analysis is focused on high-risk HPV genotypes. According to my opinion the title of the manuscript should be modified, thus emphasizing the high-risk HPV genotypes and persistent high-risk HPV infection.

 Reply: Thank you for your good suggestion, because this is a preliminary study of infection HPV viral load and methylation in HPV persistent women, due to the limited number of cases, it is difficult to explore the correlation between viral load and methylation of each HPV genotype. We will expand the sample size in the near future and show it as the second article.

Round 2

Reviewer 1 Report

Comments and Suggestions for Authors

After revisions, authors have responded to the most of requests made and have revised the manuscript. Some revisions are still necessary to improve the manuscript.

 Simple Summary: Authors introduced the Simple Summary, but this part should be modified because it should not exceed 150 words.

 Abstract: The Abstract should be adjusted as indicated in the previous revision. As indicated in the template Authors should use the style of structured abstracts, but without headings. A short part of introduction should be also included.

 Background: Authors have updated the HPV genotypes classification, but this sentence should be revised because it is not so clear.

 Authors reported: “There are over 200 identified HPV types and fifteen HPV types were classified as high-risk types(16, 18, 31, 33, 35, 39, 45, 51, 52, 56, 58, 59, 68, 73, and 82); 3 were classified as probable high-risk types (26, 53, and 66); and 12 were classified as low-risk types (6, 11, 40, 42, 43, 44, 54, 61, 70, 72, 81, and CP6108)[3].” This is the classification suggested by Munos et al in 2003 as indicated in the reference.

In 2012, the IARC published the IARC monographs on the evaluation of carcinogenic risks to humans, Vol. 100B: Biological agents. Lyon, France: International Agency for Research on Cancer; 2012. This reference is also included in the World Health Organization. WHO guideline for screening and treatment of cervical pre-cancer lesions for cervical cancer prevention, edition second 2021. In this monograph is reported that 12 genotypes are considered carcinogenic for humans (Group 1) by International Agency for Research of Cancer (IARC). HPV66 and 68 are considered possible/probable carcinogenic for human (Group 2A/B).

 Materials and Methods: The Figure 1 should be revised. 23 should be replaced with 2. Two cases were removed due to high cytological risk (ASC-H, HSIL). The legend of Figure 1 should be more descriptive.

Author Response

Thank you so much for your valuable suggestions.

After revisions, authors have responded to the most of requests made and have revised the manuscript. Some revisions are still necessary to improve the manuscript.

 Simple Summary: Authors introduced the Simple Summary, but this part should be modified because it should not exceed 150 words.

Reply: Have been modified.

 Abstract: The Abstract should be adjusted as indicated in the previous revision. As indicated in the template Authors should use the style of structured abstracts, but without headings. A short part of introduction should be also included.

Reply: Have been modified.

 Background: Authors have updated the HPV genotypes classification, but this sentence should be revised because it is not so clear.

 Authors reported: “There are over 200 identified HPV types and fifteen HPV types were classified as high-risk types(16, 18, 31, 33, 35, 39, 45, 51, 52, 56, 58, 59, 68, 73, and 82); 3 were classified as probable high-risk types (26, 53, and 66); and 12 were classified as low-risk types (6, 11, 40, 42, 43, 44, 54, 61, 70, 72, 81, and CP6108)[3].” This is the classification suggested by Munos et al in 2003 as indicated in the reference.

In 2012, the IARC published the IARC monographs on the evaluation of carcinogenic risks to humans, Vol. 100B: Biological agents. Lyon, France: International Agency for Research on Cancer; 2012. This reference is also included in the World Health Organization. WHO guideline for screening and treatment of cervical pre-cancer lesions for cervical cancer prevention, edition second 2021. In this monograph is reported that 12 genotypes are considered carcinogenic for humans (Group 1) by International Agency for Research of Cancer (IARC). HPV66 and 68 are considered possible/probable carcinogenic for human (Group 2A/B).

Reply: Thank you for your advice, have been modified.

 Materials and Methods: The Figure 1 should be revised. 23 should be replaced with 2. Two cases were removed due to high cytological risk (ASC-H, HSIL). The legend of Figure 1 should be more descriptive.

Reply: Sorry for the type error, have been modified.

Reviewer 3 Report

Comments and Suggestions for Authors

The authors have addressed all of my concerns. The revised version of the manuscript is suitable for publication.

Author Response

Thank you so much for your valuable suggestions.